# Community myths and misconceptions about sexual health in Tanzania: Stakeholders' views from a qualitative study in Dar es Salaam Tanzania

Gift G. Lukumay[1], Lucy R. Mgopa[1], Stella E. Mushy[1], B. R. Simon Rosser[2]*, Agnes F. Massae[1], Ever Mkonyi[2], Inari Mohammed[2], Dorkasi L. Mwakawanga[1], Maria Trent[3], James Wadley[4], Michael W. Ross[2], Zobeida Bonilla[2], Sebalda Leshabari[1†]

1 Muhimbili University of Health and Allied Sciences (MUHAS), Dar es Salaam, Tanzania, 2 University of Minnesota, School of Public Health, Minneapolis, MN, United States of America, 3 Johns Hopkins Medicine, Baltimore, MD, Washington, DC, United States of America, 4 School of Adult and Continuing Education, Lincoln University, Baltimore Pike, PA, United States of America

† Deceased.
* rosser@umn.edu

**Data Availability Statement:** This article is based on qualitative interviews with politicians, religious and community leaders, and academics conducted

## Abstract

### Introduction

Sexual and reproductive health problems are one of the top five risk factors for disability in the developing world. The rates of sexual health problems in most African countries are overwhelming, which is why HIV and other STIs are still such a challenge in sub-Saharan Africa. Talking about sex in most African countries is a taboo, leading to common myths and misconceptions that ultimately impact community sexual health.

### Methods

In this study, we conducted 11 key stakeholder individual interviews with community, religious, political, and health leaders (sexual health stakeholders) in Tanzania. Qualitative content analysis was used to analyze all the materials.

### Results

Two main categories merged from the analysis. The first category, "Ambiguities about sexual health" focused on societal and political misconceptions and identified ten myths or misconceptions common in Tanzania. Stakeholders highlighted the confusion that happens when different information about sexual health is presented from two different sources (e.g., community leaders/peers and political leaders), which leaves the community and community leaders unsure which one is reliable. The second category, "Practical dilemmas in serving clients", addressed a range of professional and religious dilemmas in addressing sexual health concerns. This included the inability of religious leaders and health care providers to provide appropriate sexual health care because of internal or external influences.

on the condition of anonymity. Because the topics covered are considered very sensitive, even politically dangerous in Tanzania, and potentially could be used against the key informants if someone identified them from the broader transcript, we feel ethically compelled to keep the transcripts private to protect the informant's identities. In response to the policy that authors must have a minimal data set available on request by an agency outside of the PI's control. All such requests should be sent to: University of Minnesota Institutional Review Board (attn.: Mr. Jeffrey Perkey) McNamara Alumni Center - Suite 350-2, 200 Oak St SE Minneapolis, MN 55455 Email: irb@umn.edu; ph: 612-626-5654.

**Funding:** This work received support from the following sources: Eunice Kennedy Shriver National Institute of Child Health and Human Development (NICHD), National Institutes of Health, Grant number: 1 R01 HD092655, awarded to BRSR and DM. The content is solely the responsibility of the authors and does not necessarily represent the official views of the National Institutes of Health. SpeeDx LLC (https://plexpcr.com/) provided support for this study in the form of research supplies, received by MT, through a material transfer agreement with Johns Hopkins University. The funders had no role in study design, data collection and analysis, decision to publish, or preparation of the manuscript.

**Competing interests:** The authors have read the journal's policy and have the following competing interests to declare: MT received research supplies from SpeeDx LLC (https://plexpcr.com/) through a material transfer agreement with Johns Hopkins University, and serves as a consultant to the American Academy of Pediatrics and the Church and Dwight, Inc for the Trojan Sexual Health Advisory Council for unrelated work. This does not alter our adherence to PLOS ONE policies on sharing data and materials. There are no patents, products in development or marketed products associated with this research to declare. The other authors declare no competing interests.

## Conclusion

Myths and misconceptions surrounding sexual health can prevent communities from adequately addressing sexual health concerns, and make it more difficult for healthcare providers to comfortably provide sexual health care to patients and communities. Stakeholders affirmed a need to develop a sexual health curriculum for medical, nursing and midwifery students because of the lack of education in this area. Such a curriculum needs to address nine common myths which were identified through the interviews.

## Introduction

Sexual and Reproductive Health (SRH) is "a state of physical, emotional, mental and social well-being in relation to sexuality, and not merely the absence of disease, dysfunction or infirmity"[1]. SRH is one of the top five risk factors for morbidity in the developing world [2]. The rates of sexual health problems in most African countries are overwhelming, which is why HIV and other sexually transmissible infections (STIs) remain such a challenge in sub-Saharan Africa [2, 3]. In most African countries, talking about sex is a taboo and makes many people very uncomfortable [4]. For example, parents believe that it is a teacher's duty to teach their children about sexual health at school, while teachers think that talking about sex is a basic parental responsibility. As a result, many children end up with no information regarding sexual health [4, 5]. Women and youth are the most frequent victims of sexual violence and suffer a great deal of sexual health consequences [4, 5]. Across Africa, the reasons why sexual health problems persist are complex but include lack of sexual health education, taboos against talking about sex, and socio-cultural factors such as living in a male-dominated society [6–8]

The World Health Organization (WHO) has identified gaps in sexual health care and in its current five-year strategic plan, has prioritized actions for stemming sexually transmitted infections (STIs), especially in the most affected and vulnerable countries [2]. Across the continent, there are severe consequences from inadequate (or poor) sexual health care, including high rates of infertility, abortion, sexual dysfunction, HIV/ STIs, and long-term physical and mental sequelae (e.g., increased rates of certain cancers, depression and neurosyphilis) [5, 8–10]. While these consequences affect all countries, their impact has been observed particularly in some sub-Saharan African countries where people have myths and misconceptions about sexual health [7, 8, 11–14].

Improving sexual health in Tanzania is challenging. Myths and misconceptions related to sexual health are not limited to the general population but also are common in health care providers [15–17]. For example, pregnant girls and women may refuse to use modern family planning methods, believing that doing so will lead to "watery" vaginas, infertility, and other bad side effects [15]. Similarly, healthcare providers have mixed perceptions on whether to provide family planning to a 14 years girl or not, in case they are perceived to be encouraging sexual behavior [17].

In the absence of education, myths and misconceptions are fostered and maintained by community unawareness and incorrect knowledge of sexual health [6, 12, 13, 18]. Health professionals, who the public expects to have a good understanding of sexual health (or at least sufficient knowledge to educate others), may hold the same myths and misconceptions that they grew up within a particular society [19–21]. As long as health professionals remain within their culture, given the taboos around talking about sex, their beliefs are likely to go unchallenged [16, 20–22]. Thus, health providers may pass on and reinforce inaccurate information

to patients and promote problematic care [16, 17]. Political, religious and cultural factors about sexual health play such a significant role in many African countries that these factors also perpetuate myths and misconceptions in various communities and become resistant to change [23–25].

Comprehensive sexual health education is an essential component of all health professionals' training, and a critical step in ensuring that healthcare professionals and the entire society can confront myths and misconceptions related to sexual health [26]. Educating health professionals will improve both the care patients with sexual health problems receive, as well as dissemination of accurate sexual health information through education [27]. Well-educated healthcare providers can confront the negative attitudes, myths and misconceptions related to sexual health found within their communities. However, comprehensive evidence-based sexual health education is not a component of medical, nursing and midwifery curricula, currently. To address the health concerns of Africa, there is an urgent need to develop sexual health care curricula that will address the sexual health crises impacting our continent [28].

In 2016, we conducted a 4-day pilot workshop on a sexual health curriculum with nursing and midwifery students at the Muhimbili University of Health and Allied Sciences (MUHAS) in Dar es Salaam, Tanzania. This pilot study adapted a PAHO/WHO sexual health curriculum training for Health Care Providers (HCPs) and tested it in Tanzania. The evaluation showed the sexual health curriculum was acceptable and feasible [28], but it also identified a need to tailor this curriculum to be African-centric in focus in order to maximize its effectiveness. As part of a formative research phase of study to identify the sexual healthcare needs of Tanzania, we conducted key stakeholder interviews as well as focus group discussions with midwifery, nursing and medical students and experienced health professionals. In this paper, we report the results from key stakeholder interviews with cultural, religious, political and community non-governmental organizations (NGO) leaders in Tanzania about common sexual health misconceptions related to sexual health in their jurisdictions. Identifying key myths and misconceptions about sexual healthcare is a critical step in developing an Afro-centric tailored sexual health curriculum. For this study, we define "myth" both as a traditional or widely held but false belief, idea, or practice and "misconception" as a view or opinion that is inaccurate or incorrect based on clinical evidence.

## Methods

### Study design

The study used a qualitative exploratory design in which we conducted individual interviews with key stakeholders knowledgeable about the sexual health concerns of their communities and of Tanzanian society.

**Study setting.** Data were collected from June to August 2019 in Dar es Salaam, Tanzania. Dar es Salaam was the logical choice for the site of study, given the sexual health curriculum being tailored was for medical, nursing and midwifery students at the Muhimbili University of Health and Allied Sciences (MUHAS), which is located in Dar es Salaam. MUHAS is a leading health university in East Africa and students attend from all parts of Tanzania and beyond. This study is a collaboration between two universities and was conducted under the oversight of the Institutional Review Boards, of the Muhimbili University of Health and Allied Sciences (MUHAS) in Dar es Salaam, the University of Minnesota in the US, and the [Tanzania] National Institute of Medical Research, all of whom deemed the study as exempt from human subjects review since the focus of the interviews was on clinical practice and cultural perceptions (Common Rule Exemption Category 2)."

*Sampling and data collection process.* Eleven key sexual health stakeholders were interviewed for the current study. Three were religious leaders (one imam, one priest and one

pastor), three experts in sexual health, two community leaders from (NGOs), two cultural experts and one politician. Eight of the stakeholders were male and three were female, aged 40 to 76 years, with between 8 and 43 years of experience working in their area of specialty.

Purposeful sampling was implemented to identify key stakeholders for the study. The study team identified health, cultural, political and religious leaders who were known to deal with sexual health issues. Stakeholders were contacted with an overview of the study and study purpose and invited to participate. All stakeholders contacted for the study agreed to participate and were interviewed either on the MUHAS campus or at the stakeholders' office (whichever they preferred). All interviews were undertaken in a private room to ensure convenient access and privacy. This allowed the team to capture stakeholders' experiences, and to identify the training needs of current students in each respective discipline.

In-depth interviews were chosen as an effective method to explore, in-depth, the interviewee's information, perceptions and experience of the topics under study. Each interview lasted between 45 to 60 minutes.

The interviews were conducted by three bilingual interviewers (co-authors: GGL, LRM, and SL) in English, Kiswahili, or a mix of both languages. The interviewers were clinical faculty in medicine and midwifery from MUHAS and ranged in age from 35 to 60 years. Each interviewer had more than ten years of training in sexual health. For each interview, the team decided in advance who would be the most appropriate person to interview each stakeholder. To promote comfort in being interviewed about sexual health, wherever possible, we matched the interviewer and stakeholder by gender, age, and other key characteristics.

At the start of the interview, the stakeholder was invited to respond in either English or Swahili, with most preferring to use English. Before commencing the interview, the interviewer highlighted that the interview was voluntary, that the interviewee could discontinue at any time, that all responses would be kept confidential (meaning not attributed to them personally), and that the session was being audiotaped. Stakeholders reviewed the consent form and written informed consent was provided. Each stakeholder also completed a short demographics form before commencing the interview.

*The interview guide [see attached file] consisted of nine questions that addressed three broad areas*. Sexual health problem identification, unmet sexual health needs, and the impact of sexual health policies in Tanzania. The semi-structured interview was formulated to cover multiple areas, and tailored to the expert's area of expertise. This paper reports the responses to the questions about common myths and misconceptions about sexual health.

We interviewed a minimum of two individuals from each category of stakeholders except for the politician category. Stakeholders were chosen based on many years of experience working in their roles. This likely meant they were more informed about the sexual health in their jurisdiction, knew their communities well, and so, could identify common myths and misconceptions. Data collected from each group and across all stakeholders yielded similar findings. After analyzing the transcripts, the coding team did not encounter new themes or codes, therefore, reaching saturation. We were unable to confirm saturation in the political group as we interviewed the only politician who was available.

**Data management and analysis.**   After each interview, audio files were labelled with the interview ID and uploaded onto a secure server. The audio files were transcribed verbatim and then translated into English (where necessary). In addition, notes taken during the interview were typed on a shared password-protected drive.

*Coding approach*. Both inductive and deductive codes were derived to develop the codebook and code the transcripts. A team-based approach to coding and codebook development was implemented [29, 30], beginning with open coding and followed by two coding cycles [29]. This approach to data analysis enabled a process of debate and reflection that generate a

richer understanding of the stakeholder's experiences in serving their community, in relation to sexual health.

*Coding team*. The coding and data analysis team were composed of health professionals currently practicing in their respective fields and conducting research. These providers contributed their experiences as practitioners and researchers to the team enriching the data analysis process, and serving as important member checks during the development of code definitions and data interpretations.

*Codebook*. Codebook development was an iterative process that was initiated with the creation of a list of deductive codes derived from the interview guide. Both deductive and inductive codes continued to be generated and added to the codebook iteratively during the three phases of coding. This was accomplished via regular team meetings and the use of a Google Doc where coders engaged in a process of continuous written feedback and updates to the codes and definitions that led to the reconciliation of disagreements and refinement of the codebook.

*Open coding/discovery of themes*. Open coding was conducted during a regular meeting with the team of interviewers. Open coding involved reading several times four hard-copy transcripts of the interviews and coding the interviews manually. During this meeting, the team identified early ideas and emerging themes using an open-ended coding format, without predetermined codes or categories. Pens, color highlighters, sticky notes, and large flip charts were used during this stage to gather the early ideas and organize the codes under emerging themes. Teams were organized in groups of two coders and specific questions were assigned to the team based on expertise, background, and interests. Each group manually coded the transcripts individually and compared notes and codes at the end of the open coding stage. Teams had an 80% or higher initial agreement. Codes generated during the open coding were entered into the main codebook.

The team members who conducted the interview continued with the coding. Each team coded questions pertaining to these areas of expertise and interests. All responses to questions in those topics were assigned to each sub-team. Each team coded between 2–3 questions across all interviews. Each sub-team met regularly to consult on the meaning of codes, reconcile differences, reach agreements, and update the codebook on the team document housed on the shared drive.

*First coding cycle*. During the first coding cycle, the team coded all transcripts. A holistic/segmentation process was followed to code the transcripts. In this coding process, meaning was ascribed to responses by each question contained in the interview.

*Axial coding*. During axial coding, each sub-team generated larger categories based on the findings of the first coding cycle. Codes were grouped accordingly under these larger findings.

## Results

All stakeholders described a strong commitment to working with the community, making sure that the people they serve understand them and vice versa. Stakeholders spoke of feeling discouraged when the community does not follow rules, regulations, or their advice. But they also reported that they find courage, and calm themselves in order to fulfil their duties of serving their particular community. Stakeholders identified an especially challenging situation is when the community does understand or "speak" the same language as the experts. This increases the risk of the community developing misconceptions and misunderstandings based on them using different terms that may convey inaccurate or additional information. For example, when an expert using term for a key population, but the community translates the key population into a derogatory term, it can change the meaning and tone of what is being discussed.

**Table 1. Category, sub-category and codes which emerged during analysis.**

| Category | subcategory | Selected Codes |
|---|---|---|
| *Ambiguities about Sexual Health* | Society Misconception | Negatives perceptions on Family Planning and vaccines<br>Sex as a ritual<br>Mislabeling of HIV patients<br>Unintentional sexual act<br>Community confusion on sexual issues |
| | Political misconceptions | Banning of useful hospital items<br>Homosexuality as a psychological problem<br>Political deception to the society |
| *Practical dilemma on serving client* | Professional misconception | Influences from donors<br>Perception towards Key Population care providers<br>Servicing Key Population viewed as a promotion |
| | Religions misconceptions | Sex topic is a taboo<br>Sex-related issues are private<br>Sexual deviations as bad characters |

From our analysis of the responses of stakeholders, two categories emerged: (1) *Ambiguities about sexual health* and (2) *Practical dilemmas in serving clients*. This first refers to myths and misconceptions that arise when different information comes from two different sources (e.g., community leaders/peers and politicians at different times). This leaves the community and community leaders unsure which one is reliable. *Practical dilemmas in serving clients* refers to the inability of leaders (religious, cultural and health care) to provide appropriate sexual health care because of internal or external influences (see Table 1).

## Category 1: Ambiguities about sexual health

Stakeholders pointed out that in addition to their primary role of informing the community on many things including sexual health issues, they also have to confront some widely held myths and misconceptions. While these can come from anywhere, stakeholders referred to misconceptions and misunderstandings coming from political leaders. These widely shared myths and misconceptions are passed down from generation to generation, or they persist because uninformed and/or misinformed people are resistant to change. Some stakeholders went further to express concern that while politicians are educated, they can use or even promote community misunderstanding for their own political benefit. Stakeholders described ambiguities as stemming from two main categories: societal and professional misconceptions.

**Societal misconceptions.** Stakeholders described different communities as having specific misconceptions related to sexual health (see Table 2). These myths and misunderstandings were based on inadequate knowledge in a particular community or society. One interviewee explained that in some communities, people have wrong perceptions about family planning methods and vaccine usage. When health professionals encourage people to use family planning or to get a vaccine, the recipients tend to refuse. Alternatively, they fear that the family planning methods or a vaccine will be harmful to them. Some may articulate that family planning or vaccines are a white racist conspiracy to weaken Africa.

*"They think family planning is not of good intention. They say they (white people) want to decrease the size of the population in sub-Saharan [Africa]." (Non-Government Organization Representative 2)*

Similarly, the distribution of human papilloma vaccine to adolescent school girls in Tanzania is not effective because parents decline administration, fearing that these vaccines will

**Table 2. Common community myths and misconceptions about sexual health in tanzania, as reported by stakeholders.**

| Myth or Misconception | Scientific or Clinical View |
|---|---|
| Family planning is an attempt by white people to decrease the size of the population in Sub-Saharan Africa | Access to modern contraception expands education opportunities for women, lowers infant mortality, and at the national level, sustains population growth and economic development [31] |
| The human papilloma vaccine causes infertility and makes girls fat and ugly. | There is no causal link between HPV vaccine and infertility. A large US-based study found young girls with obesity were less likely to have been HPV vaccine than normal weight girls [32] |
| Family planning is against some religion, and national interest | Tanzania has one of the highest rates of teen pregnancy and unplanned pregnancies in the world. An estimated 13.4% of pregnancies in TZ are unplanned and unwanted [33] |
| Traditional healers need to have sex with young patients for the treatment to be effective | In western societies, this would be viewed as sexual abuse and is illegal [34] |
| In some tribal cultures, parents and elders play with a boy's penis to socialize the boy sexually. | In western societies, this would be viewed as sexual abuse and is illegal [35] |
| In some tribal cultures, parents and elders pull the labia and/or clitoris of young girls to make them grow and socialize the girl sexually | In western societies, this would be viewed as sexual abuse and is illegal. Labial elongation is an effective minimal-risk technique [35] |
| Educating a boy about sex will make him want to do it. | A review of 87 studies found sex education did not lead to earlier sexual activity. A third of programs delayed sexual debut, decreased frequency of sex, and number of sexual partners [26] |
| People who are HIV positive should not have sex or children in case they transmit the virus. | HIV-positive persons with undetectable viral load cannot transmit HIV including to sex partners and have <1% transmission during pregnancy, labour and delivery [36] |
| Homosexuality is a disorder that can be treated | Homosexuality was removed as a diagnostic disorder in the DSM in 1973 and from the ICD-10 in 1990. There are no evidence-based effective treatments to change a person's sexual orientation [37] |

cause infertility in their daughters and because they do not believe the vaccine will prevent cervical cancer.

*"As usual in some African countries, you find parents are refusing to give consent and say that there is some hidden agenda behind the vaccine. They don't believe that it is there just to protect their children from cancer. . . Well, they say, that we (the vaccine providers) want to stop their girls from getting children, aaah sterilizing them." (Non-Government Organization Representative 2)*

However, in some cases, community members can become confused when the information they research from their health professional conflicts with the information they head from politicians and other leaders.

*"I have a client whom I was counselling for family planning. Then, the client was like you (health professionals) are telling us to use family planning methods, but our president said we should bear kids because education is free." (Non-Government Organization Representative 1)*

While there is an assumption that sex is voluntary, stakeholders said that this is not always true in some African cultures. A sexual act may be misinterpreted and may carry different

meanings within a particular society. For instance, some traditional healers would require either their patients or young boys and girls to have sex with them as a means of treating an illness.

*"If you go to a traditional healer and [he] tells you must go have sex with him or a young girl or a young boy, for you that's tricky. For me, that is sex and a ritual . . . parents play around with the penis of the young boys, and then they say, ahaaa, and the boy would laugh, in fact, and that. . .that. . .when I now think about it that was sex socialization." (Cultural expert 2)*

In other African cultures, in order for a parent to ascertain his son's manhood or sexual arousal, a child's parent may rub his son's penis and observe the young boy's reaction. This is considered normal in some African cultures.

*"These are cultures which exist in this country, where we have the elderly when they are socializing with young girls, they have means where one tribe pulls the labia so that they become big, the others pull the clitoris until when it looks like a little penis" (Cultural expert 2).*

Given the taboo of talking about sex, and at the family level, the absence of parental sex education, it is difficult for young people to know what is right or wrong. As one stakeholder explained, a man may rub himself against a woman's behind to the point of ejaculation in a congested public place like in a bus or at the market without her being aware. Without education that this is not normal, and that frotteurism is diagnostic of paraphilia, the youth may end up either with others minimizing his behavior or being taken into custody instead of being referred for assessment and treatment.

*"For example, when you talk about parent-youth communication, when they want to talk about sexual health-related issues, so this norm of not talking is a misconception . . . It is really difficult for a father [in Tanzanian culture] to start talking with his child, daughter, about sex so it is a part of that challenge." (Health Expert 2)*

A public health expert emphasized misconceptions regarding HIV patients. Some community members maintain a misconception that once someone is infected with HIV they cannot get married to anyone or bear children because they will infect the partner or the baby. In this era of antiretroviral therapy (ARV), women who are on successful on ARV treatment can be married and have children without infecting their partner.

*"People need to know that even people who are HIV positive, once they are on treatment, their viral load is suppressed and they can get married and they can have babies who are very fine. So, that is something regarding sexual health which I think they are still some misconceptions and misunderstanding." (Health Expert 3)*

**Political misconceptions.** Politics does have its contribution towards sexual health misconceptions, which affects the community. Stakeholders described a tendency where politicians use community misunderstandings, or the reserved culture, to advance their agenda. Stakeholders pointed out that there are occasions where political representatives contribute to community misunderstanding towards HIV prevention. For example, the Tanzanian government banned the sale of K-Y gel lubricant by pharmacies. The stated reason behind the ban was key populations (i.e., men who have sex with men) use lubricant. The politician's stated concern was that the lubricant violates the law by facilitating same sex relations which are illegal. But stakeholders noted that lubricants can also be used by heterosexuals, women may need

them for gynecological exams, and lubricant reduces friction during sexual intercourse for menopausal women and lowers the risk of HIV transmission.

*"For example, a lubricant such as K-Y gel, they have to buy from a pharmacy or whatever and once upon [a time] it was forbidden by the ministry. Is that not a misunderstanding that we want to stop transmission of HIV and then you, as a boss, you stop using this, and this is one of the tools that one has to use to minimize the friction, sometimes the rate of transmission of HIV." (Health Expert 2).*

Stakeholders stated that many politicians avoid speaking about the actual reality of sexual health to the communities. Most politicians were perceived as pretending to be against key populations such as sex workers or men who have sex with men and overstate their contribution to HIV epidemic in the country caused by having multiple sex partners while minimizing heterosexual risk behaviors.

*"They [are] pretending that they are against sex workers, but yet sex workers are really few here in Dar es Salaam during parliamentary proceedings in Dodoma (the capital city)." (Cultural Expert 2)*

Stakeholders said some politicians in Tanzania use the excuse that sex education might influence innocent youth to practice such behaviors. While politicians are supposed to be role models, they deny or stigmatize some behaviors while engaging in them privately. Stakeholders saw this as a deception to the community because it makes the politician look good while depriving the community of receiving important information regarding sexual health.

*"So, with the politicians, . . . many of them will oppose the very behaviors they practice, and particularly because the practices are private and no one will know. But when they are in public, that is the language they speak which is different." (Cultural expert 2)*

Some politicians advocate that people who are homosexual have psychological problems and should be given medical or psychological help to become normal (i.e., heterosexual), while others advocate that since they choose to practice same sex relations, then they should be punished.

*"There are people who believe that there are people who are born like that, with those maybe I would say physiological or psychological impairment hence attracted to same sex. . . or if they have decided to be like this, they need to be punished because this is something which is against our community as well as our laws. . . But what I know is with a proper education they can become good people with acceptable behaviors. But mind you it needs time to see the impact." (Political leader)*

## Category 2: Practical dilemmas

Stakeholders reported some practical challenges and dilemmas in addressing sexual health concerns in Tanzania. Stakeholders noted that because politicians have public power, they can announce misinformation regarding sexual health. Regardless of the truth of what they assert, it becomes difficult to change the public statement which has already been accepted by the people. Cultural experts claimed that most of the sexual health services in Tanzania are donor oriented. Therefore, they are fund driven and difficult to query their effectiveness. Religious leaders saw themselves as committed to preaching the word of God primarily and not about

advocating for sexual health (regardless of how much they know about it). There were two types of practical dilemmas stakeholders encountered: professional misconceptions and religious-based misconceptions.

**Professional misconception.** Stakeholders reported that conflicts emerge when professionals claim one reality when the community members experience another. For example, in family planning, the professionals are vulnerable to believing the science that a medication tested elsewhere in the world is fine, while the community members are reporting side effects. This can lead to the professionals minimizing the community's experience.

> *"But, here, the science will say there are no problems but users had seen those problems, so for you coming from MUHAS (i.e., the Health Sciences University), what you hear from these individuals those are misconceptions. But these are reality in their own world and you're the one who has misconceptions about what they are saying." (Cultural Expert 1)*

In less developed countries like Tanzania, most of the health services are donor funded. When members of the community report problems with the inefficiency of a particular service or the quality of the item donated to them (e.g., mosquito nets), the health professionals may deny, minimize or disregard the concern raised by the community and condemn them for expressing concern. Instead, health professionals support the donors who supply the items and the funds because of the risk of destroying the political or professional relationships and losing the funding. Health professionals must be accountable, however, to research and confirm the communities' concerns to better engage sponsors about the needs of the community.

> *"Yeah, politicians are the results of science. Because even if I will go and ask for money for doing research, if insect-treated nets are reducing manpower, they won't give it. They will end up saying WHO [the World Health Organization] they have already said that this is safe, [so] who are you by the way? Do you want to reinvent the wheel? WHO, they have already [said] its safe for malaria prevention, man power for what while people are dying of malaria? But again, they are pushed by funders too, and they need that fund [so] they have no option. And when they bring us, we have to take it to the intended area." (Cultural Expert 1)*

There are several myths and misconceptions associated with high risk or key populations (i.e., sex workers, men who have sex with men, transgender persons, prisoners, and individuals with substance use disorders). First, both the community and health professionals may hold significant misgivings about what is appropriate sexual health care for these populations. For example, they may worry that providing condoms may encourage behavior they disapprove of. Second, health-care providers face being stigmatized if they become known for treating key populations. Community members and other health care providers may wonder if providers who treat men who have sex with men are gay themselves or engage in homosexual behavior. This is guilt by association and can lead to discrimination and segregation from other care providers. Similarly, if a politician supports human rights and equal health care services to all, they may be (mis)perceived and labelled as an activist by the community and by opposition politicians rather than as someone interested in the promotion of quality care for all to better the health of the overall population.

> *"The other misconceptions are if society sees you working with key populations (KPs), they consider you as one of the KP also. So, even in health care facilities other health works, start to segregate you... Even within the government, there are people who would view us as puppets, and others say we are fighting for KPs rights because of donors only and they even went further and say we are being paid to promote"... (Political leader)*

**Religious misconceptions.**    Religious leaders also identified barriers and misconceptions regarding sexual health in those they serve. Regardless of the religion or denomination, they said discussion about sexual health topics in churches or mosques was considered shameful and taboo. Religious stakeholders considered sex a private matter while acknowledging the importance of talking about it. Many sexual health issues were considered unmentionable because of the belief that sex is a sacred act reserved for marriage. This view was not only held by the religious stakeholders but also, in their experience, by their congregations. In addition, behaviors like same sex relationships and anal sex were highly regarded as immoral behaviors which were against their religious values.

*"People think that this (sex) is something which is so private, very private you cannot speak about it. So sacred, we do not need to talk about it . . . We do not say it [talk about it] at church and parents at home also they do not say it. . . We do not talk about sex of the same sex. . . It is an abomination in our context but this is something people are doing and it has effects." (Religious leader 2)*

Stakeholders acknowledged that many sexual practices and health concerns are common in our society, but people do not talk about them. Hence, no help is provided to those with the concern They acknowledged the role of religious leaders in confronting the silence around sexual health concerns.

*"I have encountered some cases whereby there was a religious school and people are complaining that kids are introduced issues related to anal sex by either staffs, or other peers from suburbs. You find children are congested together in rooms. . .. So, we need to talk about these issues. Because we as nation there is nobody who is exempted when you are talking about sex, these staffs cover everybody and we need to help." (Religious leader 3)*

## Discussion

This study identified ten common myths and misconceptions viewed by stakeholders related to sexual health in Tanzania (see Table 2). These findings provide insight into what sexual health myths and misconceptions stakeholders encounter when serving their communities. These can be used to identify critical knowledge gaps that need to be addressed when developing a sexual health curriculum in Tanzania. Our findings indicate that there are several common myths and misconceptions that are prevalent among community members across different jurisdictions.

Myths and misconceptions which were raised by stakeholders included negative perceptions on family planning, mistrust of health care providers who treat key populations, vaccines especially the effects of the HPV on fertility, infectivity of people living with HIV, what is sexual health, and what things can be treated. Across their varying expertise, all the experts identified the cultural taboo about talking about sexual issues/topics as a major issue.

With regards to family planning, experts said community members perceived greater health risks of using modern family planning methods than the evidence suggests, believing that they are more harmful than as described in educational leaflets or based on health care providers' advice [4, 6, 8, 19]. Community members may often believe that there is a hidden agenda to promote infertility, population control, and weight gain. This is despite the documented health benefits associated with the use of modern contraceptives [10, 31, 38]. These misconceptions leave women of reproductive age without effective methods of pregnancy prevention and increase the risk of unplanned pregnancies. And they are consistent with the results of another

study recently conducted in Dar es Salaam with young women. Participants in that study reported that they fear using family planning because they will end up having vagina discharge, watery vaginas, uterine tumors, cervical cancer, as well as infertility [15].

The benefits of HPV vaccination for cervical cancer and anogenital wart prevention have been documented scientifically and it has been proved safe for use [39, 40]. Stakeholders stated that community members perceive HPV vaccination as a strategy to depopulate their community by sterilizing adolescent girls. Low uptake of HPV vaccination will ultimately result in needlessly high rates of cervical cancer and other anogenital conditions [40]. In addition, these perspectives lead to mistrust between care providers and their clients due to the introduction of suspicion into the patient-provider relationship. This finding is consistent with studies in other countries that have documented the association between such myths and misconceptions about the safety of the HPV vaccine and suboptimal vaccination rates [11, 18].

Additionally, in this era of effective antiretroviral (ARV) treatment, the viral load of individuals living with HIV can be suppressed to the point where HIV may not be transmitted during pregnancy, childbirth, or sexual intercourse [9, 41]. Despite this, community members were described as continuing to have negative perceptions toward people living with HIV and exaggerated risk of HIV transmission to partners and children even with effective treatment. Similar misconceptions have been reported in other countries [7, 12, 13, 42]. Such misconceptions may infringe on the legal rights of individuals living with HIV to marry and parent.

Misconceptions about and stigmatization of key populations (especially men who have sex with men) were common, severe and pervasive [43–45]. Stakeholders identified other healthcare providers as stigmatizing colleagues who provide care to key population patients. High-quality, evidence-based care was reframed as promoting same sex sexual behavior or advocacy. Healthcare workers who provide high-quality care regardless of patient background were viewed with suspicion. These findings are consistent with research conducted in Jamaica and the Bahamas [46]. All healthcare workers take an oath to provide treatment without prejudice, therefore, the inter-collegial suspicion and stigmatization noted by stakeholders in this study are especially concerning. This toxic and unprofessional behavior perpetuates suboptimal care and may lower morale in the healthcare workforce.

In addition to the usual sex health challenges that exist in all or almost all countries, stakeholders raised the challenge of ritual sexual practices in some tribes and cultures. This includes sex as a treatment between traditional healers and patients, and also genital touching between older relatives and children. Traditional healers' sexual practices, especially intercourse with patients, are both an HIV/STI concern, and a form of sexual exploitation [47].

Myths and misconceptions are major reasons community member do not follow medical advice across several areas [7, 11, 12, 14, 18, 19, 42]. This was thought to emerge when community leaders promote information that is counter to evidence-based medical advice. This results in confusion within the community as to what information to believe. Several stakeholders described scenarios in which healthcare providers have advised the community based on scientific evidence [27, 28], but this information contradicts the views of some politicians. Politicians were often described as deliberately promoting misinformation in some cases to advance their careers. Hypocrisy also emerged as an issue given that some politicians publicly denouncing some sexual behaviors as immoral while privately engaging in them [19].

Lastly, churches, masjids, and other houses of worship where the community gathers are potential places to promote accurate sexual education, positive attitudes, and open discussion. However, religious leaders confirmed that they consider sex as a private matter that should not be spoken about publicly. This perspective was present despite leaders knowing that their members have or may be at risk of a number of sexual problems. There was also a fear expressed by some stakeholders that talking about a sexual issue may encourage others,

especially the young, to experiment. In religious groups, the taboo/culture of not talking about sex [23–25], prevents open discussion of sexual health and maintains sexual misconceptions.

In many countries, there are misconceptions regarding sexual health but they may be exacerbated in low-income countries [6, 7]. Lack of education in general, and lack of access to accurate sexual information in particular, likely maintain all of the myths and misconceptions identified by the stakeholders. Encountering providers who hold these beliefs or are unprepared to counteract them may deter patients who have sexual health problems from seeking care. Without high-quality sexual health advice and care, patients may engage in sexual risk-taking behaviors that result in poor health outcomes including high rates of unwanted pregnancies, HIV/STI, sexual exploitation and abuse. To dispel these myths and misconceptions, it is important to provide comprehensive evidence-based sexual health and sexuality education for health professionals. In the Tanzanian context, our results indicate that sexual health education should specifically address the effectiveness and benefits of family planning, HPV vaccination, ARV treatment, and non-judgmental practices of providers towards clients and colleagues who serve key populations at risk for sexual health disparities.

In most high-income countries, medical, nursing and midwifery students have access to sexual health education as part of their professional training [48], which is not the case in most African countries. To be effective in Africa, such a curriculum needs to address the issues identified by the stakeholders and potentially other unidentified areas to ensure a comprehensive education for clinical practice. Students and practicing professionals need training so that they can learn accurate information about sexual behavior and clinical health issues, how to address sexual health problems using evidence-based interventions, and how to provide sexual education and services within the community. Once such a health curriculum is evaluated, broad dissemination to include professional training programs and continuing education should be a priority.

We expect that after healthcare providers are trained in sexual health, they will begin to educate their patients during clinic visits and through community education. Social support, and Train the Trainer programs among the community workers will be a potential adjunct to create broad knowledge within the community given that myths and misconceptions have a negative impact on individual and community health [49–51].

## Limitations of the study

In this study, we interviewed diverse key stakeholders and community leaders trying to understand common misconceptions and misunderstandings related to sexual health in different communities in Tanzania. While we had interviews with multiple community, religious, cultural and sexual health experts, only one politician was willing to be interviewed. Eleven interviews were conducted for the present study, which might seem like a small sample. However, during the last three interviews, the same issues emerged repeatedly suggesting saturation was being reached. While the interviews were conducted in both Swahili and English, the scripts were translated and analyzed in English. We acknowledge that some of the meanings of the stakeholders might have been lost in the translation process. To minimize this, six of the authors, all bilingual in English and Kiswahili, were involved in the analysis.

## Directions for future research

The purpose of this study was to identify commonly held myths and misconceptions in Tanzania so as to inform a tailored sexual health curriculum for healthcare students in Dar es Salaam. The logical next step is to embed these findings into the curriculum and test whether education can modify these beliefs. Such a trial is underway. Beyond sexual health education

for healthcare students, these findings have wider implications for community health education and policy regarding sex education in schools. In undertaking such initiatives, challenges include how to be respectful of Tanzanian culture and mores, while also promoting and advocating for better sexual health. Given the diversity of Tanzania's culture, future research should also identify regional, tribal and demographic (e.g., gender, religious, rural-urban and age) differences where local communities and individuals may hold different myths and misconceptions.

## Conclusions and recommendations

This study has identified several key sexual health myths and misconceptions that function as a barrier to clients and the community seeking sexual health care. These myths and misconceptions also create a barrier to health care providers providing quality sexual health care, especially stigmatized and vulnerable patient groups. Healthcare providers need to be prepared to challenge these myths and misconceptions but may have inadequate sexual health training to respond when challenged by community members. An evidence-based comprehensive sexual health curriculum for medical, nursing, and midwifery students is needed to train health professionals with accurate, evidenced-based information to treat sexual health concerns. This curriculum in Tanzania should specifically address the common myths and misconceptions identified by the stakeholders in this paper and provide opportunities to practice key communication skills for optimal clinical effectiveness and community advocacy.

## Supporting information

**S1 Checklist. COREQ checklist.**
(PDF)

**S1 File.**
(DOCX)

**S2 File. Code book for sexual health myths and misconception.**
(DOCX)

**S3 File. Key informant guide.**
(DOCX)

## Acknowledgments

The paper is dedicated to co-Principal Investigator, Dr. Sebalda Leshabari who died during the course of this study.

## Author Contributions

**Conceptualization:** Gift G. Lukumay.

**Funding acquisition:** B. R. Simon Rosser, Sebalda Leshabari.

**Investigation:** Gift G. Lukumay, Lucy R. Mgopa, Ever Mkonyi, Sebalda Leshabari.

**Methodology:** Gift G. Lukumay, B. R. Simon Rosser, Zobeida Bonilla.

**Supervision:** B. R. Simon Rosser, Michael W. Ross, Sebalda Leshabari.

**Writing – original draft:** Gift G. Lukumay.

**Writing – review & editing:** Lucy R. Mgopa, Stella E. Mushy, B. R. Simon Rosser, Agnes F. Massae, Inari Mohammed, Dorkasi L. Mwakawanga, Maria Trent, James Wadley, Michael W. Ross.

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
