## [Decision Letter · Decision Letter 0]

14 May 2021

PONE-D-20-40972

Community Myths and Misconceptions about Sexual Health Promotion in Tanzania: Stakeholders’ Views from a Qualitative Study in Dar es Salaam Tanzania

PLOS ONE

Dear Dr. Rosser,

Thank you for submitting your manuscript to PLOS ONE. After careful consideration, we feel that it has merit but does not fully meet PLOS ONE’s publication criteria as it currently stands. Therefore, we invite you to submit a revised version of the manuscript that addresses the points raised during the review process.

There was divergence in the views of the reviewers. Please address all comments made by reviewer 1, including the use of the COREQ checklist. 

Regarding reviewer 2's comments, both below and in the attached - please address all comments, with two exceptions: 

1. Sample size and saturation. Kindly note reviewer 2's feedback on saturation and carefully consider how this impacts your study design and analysis. In addition, please provide further detail on saturation reached in the data collection, sampling or analytical stages. You could refer to Saunders et al 2018 (https://www.ncbi.nlm.nih.gov/pmc/articles/PMC5993836/). 

2. Regarding reviewer 2's comments on the need to *'To know both about sexual health and associated health problem, the primary stakeholder should be the targeted community (e.g, youth, adolescent, and people with the problem) which you didn't include as a study participant.*'  Please address this by providing a clear and specific objective of the study and the approach to interviewing the stakeholders that were included in the study. 

We look forward to receiving your revised manuscript.

Kind regards,

Caroline Anita Lynch

Academic Editor

PLOS ONE

Journal Requirements:

2)  Please include additional information regarding the interview guide used in the study and ensure that you have provided sufficient details that others could replicate the analyses. For instance, if you developed an interview guide as part of this study and it is not under a copyright more restrictive than CC-BY, please include a copy, in both the original language and English, as Supporting Information.

3) Thank you for including your ethics statement:  "University of Minnesota institutional review boards (IRB) Case Study: Study00004044, as well as the National Institute for Medical Research in Tanzania.".   

a).  Please amend your current ethics statement to confirm that your named institutional review board or ethics committee specifically approved this study.

b).  Please provide additional details regarding participant consent. In the ethics statement in the Methods and online submission information, please ensure that you have specified what type you obtained (for instance, written or verbal, and if verbal, how it was documented and witnessed). If your study included minors, state whether you obtained consent from parents or guardians. If the need for consent was waived by the ethics committee, please include this information.

4)  We note that you have indicated that data from this study are available upon request. PLOS only allows data to be available upon request if there are legal or ethical restrictions on sharing data publicly. For more information on unacceptable data access restrictions, please see http://journals.plos.org/plosone/s/data-availability#loc-unacceptable-data-access-restrictions.

Reviewers' comments:

Reviewer's Responses to Questions

**Comments to the Author**

1. Is the manuscript technically sound, and do the data support the conclusions?

Reviewer #1: Yes

Reviewer #2: No

2. Has the statistical analysis been performed appropriately and rigorously? 

Reviewer #1: N/A

Reviewer #2: N/A

3. Have the authors made all data underlying the findings in their manuscript fully available?

Reviewer #1: Yes

Reviewer #2: No

4. Is the manuscript presented in an intelligible fashion and written in standard English?

Reviewer #1: Yes

Reviewer #2: No

5. Review Comments to the Author

Reviewer #1: Thank you for the opportunity to review this interesting manuscript entitled Community myths and misconceptions about sexual health promotion in Tanzania: Stakeholders' views from a qualitative study in Dar es Salaam Tanzania. I read the paper with great enthusiasm as it pertains to very critical information for the SSA. I know its unbelievable that we are still struggling with myths and misconception in today's world but that is indeed our current situation and therefore I find this paper quite an important paper! it is well-written and concise, easy to read and interesting. I do however, have some comments for the authors to consider, particularly on the methods section.

Comments:

Methods

I find this section rather containing insufficient information. for example, I'm curious to know who conducted the interview with the religious leaders? this information is not described in the manuscript... what is the gender, experience, age, and other key characteristics of the interviewers???? there were religious leaders and cultural experts in this study and therefore it is important to know the background characteristics of interviewers to identify any potential bias from them...I would also recommend the use of the COREQ checklist to assist here.

the analysis piece is well described!

this section would benefit from a proper restructuring of sub-sections. I find the data collection and setting mixed up with the data collection procedures, sampling approach and some setting info all muddled together... I'd advise the authors to specifically separate the setting and sampling approach, describing how they chose/selected their stakeholders/ participants and why. then describe the process they followed in collecting data and conducting the interviews

page 6, line 122: i would remove this as it is repeated in the next page at line 125.

Results

I noted that the authors use the words "participants", "informanst", "key informants" and "stakeholders" interchangable... i'd recomment to stakeholders and use that consistently throughout the paper as that is the term more in line with the title of the manuscript..

Reviewer #2: Sexual health is a wide concept and its public health importance or related health problems are varied based on gender and age of the population. In this regard, the study lack focus at all and as well as depth.

One of the big drawbacks of this study is the insufficiency of the study participant in numbers, and representation to understand the possible myths and misconception. The public health importance of sexual health problems for young, adult, elders, and male and female are different. So defining the target population should be the primary purpose for such study. The myth and misconception on sexual health also different with related sexual and reproductive health problems. You tried to touch few issues related to sexual health such as HIV/AIDS, homosexuality, and the like but everything is very shallow and not systematic.

To know both about sexual health and associated health problem, the primary stakeholder should be the targeted community(e.g, youth, adolescent, and people with the problem) which you didn't include as a study participant.

Moreover, there is a concept called "Saturation" in the qualitative study. According this concept, the maximum sample size depends on the level of saturation. To determine whether the information you get is saturated or not, its needs to interview a minimum of 2 individual from each category of people interviewed. For example, a minimum of 2 health professionals, 2 politicians, and the like should be interviewed to say the information needed from that category is saturated or not while this study missed this concept as a single individual from each category of people considered as stakeholder interviewed (e.g. Health profession or politician) and you never talk to any category of people expected get sexual health services.

Your analysis process also simply describing the theoretical definition of coding and fail to describe what was actually done during coding, grouping, and interpretation of the finding.

6. PLOS authors have the option to publish the peer review history of their article (what does this mean?). If published, this will include your full peer review and any attached files.

Reviewer #1: **Yes: **Kim Jonas, PhD

Reviewer #2: No

---

## [Author Response · Author response to Decision Letter 0]

24 Jun 2021

3rd May 2021

To: Managing Editor

Plos One

RE: REVIEWERS RESPONSES. 

Dear Editor and Reviewers:

Thank you for your helpful comments and positive review of our manuscript entitled “Community Myths and Misconceptions about sexual health in Tanzania: Stakeholders' Views from qualitative study in Dar es Salaam Tanzania”. Below please find a point-by-point summary of the comments (in regular type) together with our response (in italics).

Editor. 

1) Please ensure that your manuscript meets PLOS ONE's style requirements, including those for file naming

Response: Both the senior author (G.G. Lukumay) and our staff person who assists in the manuscript submission (Ms. Lele) have checked the instructions to authors to ensure the manuscript conforms to PLoS One’s requirements. 

2) Please include additional information regarding the interview guide used in the study and ensure that you have provided sufficient details that others could replicate the analyses. For instance, if you developed an interview guide as part of this study and it is not under a copyright more restrictive than CC-BY, please include a copy, in both the original language and English, as Supporting Information.

The interview guide has been attached as a separate file during resubmission please check. 

3) Thank you for including your ethics statement: "University of Minnesota institutional review boards (IRB) Case Study: Study00004044, as well as the National Institute for Medical Research in Tanzania." 

a). Please amend your current ethics statement to confirm that your named institutional review board or ethics committee specifically approved this study.

Thank you for this reminder. We have amended this statement and now it reads,“ This study is a collaboration between two universities and was conducted under the oversight of the Institutional Review Boards, of the Muhimbili University of Health and Allied Sciences (MUHAS), the University of Minnesota, and the [Tanzania] National Institute of Medical Research, all of whom deemed the study as exempt from human subjects review since the focus of the interviews was on clinical practice and cultural perceptions (Common Rule Exemption Category 2).” 

b). Please provide additional details regarding participant consent. In the ethics statement in the Methods and online submission information, please ensure that you have specified what type you obtained (for instance, written or verbal, and if verbal, how it was documented and witnessed). If your study included minors, state whether you obtained consent from parents or guardians. If the need for consent was waived by the ethics committee, please include this information.

 The section has been amended to show which type of consent was obtained, now it reads “Before commencing the interview, the interviewer highlighted that the interview was voluntary, that the interviewee could discontinue at any time, that all responses would be kept confidential (meaning not attributed to them personally), and that the session was being audiotaped. Stakeholders reviewed the consent form and written informed consent was provided. Each stakeholder also completed a short demographics form prior to commencing the interview.” Please see page 6

The same text has been added to the Ethics Statement field of the submission form via edit submission.

We note that you have indicated that data from this study are available upon request. PLOS only allows data to be available upon request if there are legal or ethical restrictions on sharing data publicly.

We request restriction on ethical and legal grounds. The key informants, some of whom are prominent public figures (politicians, pastors, imamsA, CEOs of community-based organizations), agreed to participate on condition that their expert views would be anonymous. There is a risk of harm to these public figures if the transcripts were made available, for example, to the press and their comments taken out of context. Similarly, the informants would have grounds to sue the study if the press or someone opposed to their work identified the person from the transcript and used it as ammunition against them. 

Reviewer 1

Methods

I find this section rather containing insufficient information. for example, I'm curious to know who conducted the interview with the religious leaders? this information is not described in the manuscript... what is the gender, experience, age, and other key characteristics of the interviewers???? there were religious leaders and cultural experts in this study and therefore it is important to know the background characteristics of interviewers to identify any potential bias from them...I would also recommend the use of the COREQ checklist to assist here

We have included the demographic characteristics of interviewers in page 6 as requested. In addition, we have attached the COREX checklist with this resubmission as suggested by reviewer and editor. 

The analysis piece is well described! this section would benefit from a proper restructuring of sub-sections. I find the data collection and setting mixed up with the data collection procedures, sampling approach and some setting info all muddled together... I'd advise the authors to specifically separate the setting and sampling approach, describing how they chose/selected their stakeholders/ participants and why. then describe the process they followed in collecting data and conducting the interviews

Thank you for this good suggestion. We have separated subtitles in this section for easy clarity. Now we have setting, data collection and sampling as independent subtitles. 

page 6, line 122: I would remove this as it is repeated in the next page at line 125.

We have removed the repeated sentence as you suggested. 

Results

I noted that the authors use the words "participants", "informants", "key informants" and "stakeholders" interchangeable... I'd recommend to stakeholders and use that consistently throughout the paper as that is the term more in line with the title of the manuscript.

We agree, and have replaced the other synonyms with “stakeholders” as suggested by the reviewer. 

Reviewer 2

To know both about sexual health and associated health problem, the primary stakeholder should be the targeted community (e.g., youth, adolescent, and people with the problem) which you didn't include as a study participant.

The reviewer makes an excellent point, and particularly if this study had a narrow focus (e.g., on one or a few key populations), we would have done that. There were three reasons why we chose this approach. First, the aim of the study was to identify common myths and misconceptions about sexual health in Tanzania to inform a sexual health training curriculum for health students. For this reason, it made sense to conduct interviews with expert key informants (most of whom have worked for many years across multiple key populations) and community leaders to identify the common myths and misconceptions health students will encounter in their patients in Tanzania. Second, this is only one part of a larger investigation involving other methods, including focus groups with the target population (i.e., health students), focus groups with experienced health professionals, and a structural analysis of clinics and hospital systems. Third, based on our pilot curriculum and a review of sexual health training for health students, we had already included the most stigmatized key populations in Tanzania (i.e., LGBT persons, sex workers) as panelists in the curriculum. So, we were confident they could address their experience directly with the students. 

Moreover, there is a concept called "Saturation" in the qualitative study. According this concept, the maximum sample size depends on the level of saturation. To determine whether the information you get is saturated or not, its needs to interview a minimum of 2 individual from each category of people interviewed. For example, a minimum of 2 health professionals, 2 politicians, and the like should be interviewed to say the information needed from that category is saturated or not while this study missed this concept as a single individual from each category of people considered as stakeholder interviewed (e.g. Health profession or politician) and you never talk to any category of people expected get sexual health services.

We confirm our study team included a qualitative expert methodologist who trained all members in qualitative methods including the goal of saturation. Our goal was to have 2-3 experts in each category participate, and indeed we had 2 cultural leaders, 3 sexual experts, 2-community leaders, and 3 religious’ leaders. The exception was politicians where we tried to recruit multiple politicians but were only able to recruit one who was willing to discuss sexual health and key populations. This was perhaps not surprising given Tanzania is a highly conservative country and stigmatized key populations (especially men who have sex with men and sex workers) are politically highly sensitive topics for discussion. We have now included this as a limitation in the limitations section. 

Your analysis process also simply describing the theoretical definition of coding and fail to describe what was actually done during coding, grouping, and interpretation of the finding.

We confirm that the coding process, as described in page 7-8, was the process actually done. The result of the described process brought 2 Categories, 4 subcategories as well us 4 to 5 codes to every subcategory. 

Thank you to the reviewers for their helpful feedback.

Best regards,

Gift Lukumay, BScN, MSc.PHEC

Department of Community Health Nursing 

Muhimbili University of Health and Allied Sciences

---

## [Decision Letter · Decision Letter 1]

14 Dec 2021

PONE-D-20-40972R1Community Myths and Misconceptions about Sexual Health in Tanzania: Stakeholders’ Views from a Qualitative Study in Dar es Salaam, Tanzania.PLOS ONE

Dear Dr. %Simon Rosser%,

Thank you for submitting your manuscript to PLOS ONE. After careful consideration, we feel that it has merit but does not fully meet PLOS ONE’s publication criteria as it currently stands. Therefore, we invite you to submit a revised version of the manuscript that addresses the points raised during the review process.

Please pay close attention to the reviewers comment and respond to each before resubmitting.==============================

We look forward to receiving your revised manuscript.

Kind regards,

Mary Hamer Hodges, MBBS MRCP DSc

Academic Editor

PLOS ONE

Journal Requirements:

Additional Editor Comments (if provided):

Thank you for taking the reviewers comments on board. More attention to detail required for this next revision please.

Reviewers' comments:

Reviewer's Responses to Questions

**Comments to the Author**

1. If the authors have adequately addressed your comments raised in a previous round of review and you feel that this manuscript is now acceptable for publication, you may indicate that here to bypass the “Comments to the Author” section, enter your conflict of interest statement in the “Confidential to Editor” section, and submit your "Accept" recommendation.

Reviewer #1: (No Response)

Reviewer #3: (No Response)

2. Is the manuscript technically sound, and do the data support the conclusions?

Reviewer #1: Yes

Reviewer #3: Yes

3. Has the statistical analysis been performed appropriately and rigorously? 

Reviewer #1: N/A

Reviewer #3: N/A

4. Have the authors made all data underlying the findings in their manuscript fully available?

Reviewer #1: Yes

Reviewer #3: (No Response)

5. Is the manuscript presented in an intelligible fashion and written in standard English?

Reviewer #1: Yes

Reviewer #3: Yes

6. Review Comments to the Author

Reviewer #1: I'd like to thank the authors for attempting to address my and reveiwer2's previous comments. I am happy with this revision but have further suggestions...

Abstract:

the authors mentioned in their response letter that they opted for "stakeholders" but there's informants in line 34.

Introduction

Please check your referencing... only one box is needed to house the ref number... e.g., myths and misconceptions are common in Tanzania [11-15] and not [11] - [15], no, this is not how it goes... this comment applies throughout the manuscript in text referencing. if the references are not chronological you can still put them in one square box this way [5, 7,11-4]... hope this is making sense. watch out for spacing before the in text refencing/ square boxes

Methods

line 125, please rearrange to read: Sampling and data collection process

line 139, here's key informants again...?

I expected to see the reference to the "Additional file" of the interview guide in this section but I cannot find it...It would be ideally places somewhere by line 148

Nicely described analysis process!

Results

Table 2, page 14... there's a typo- ....unplanned pregnancies in the "world"

line 277, another "informant" and in line 327, 338, 348... there's more, please ensure you check this for consistency!

Otherwise, this is a nice paper and timely for health providers.

Reviewer #3: Comments to the Authors

Thanks to Plos One for the opportunity to review this manuscript and to the authors for their interesting work on this important topic. Generally, the paper is well-written. The authors did a good job in addressing the comments raised in a previous round of review. However, there are some issues in the manuscript that need to be addressed before it can be considered for publication. I have outlined these concerns below and hope that my comments are helpful to the authors if they are given the opportunity to revise the paper:

Introduction

-The introduction is fairly written. But I wonder if in Tanzania there are no studies that addressed myths and misconceptions around sexual health. I would urge the authors to acknowledge the existing evidence and situate their study within it.

-It would be good to add some lines stating the situation of sexual health problems in Tanzania and the related myths and misconceptions.

Methods

-Lines 131-133, the authors state: “The study team identified health, cultural, political and religious leaders who were known to deal with sexual health issues”. It is not clearly described who helped the researchers in identifying and approaching these key informants to participate in the interviews.

-Under the sub-section “Setting”, line 145-146, the authors state: “Data were collected from June to August 2019 in Dar es Salaam, Tanzania.” I think the authors should provide further description of the setting of this study. This is particularly important for qualitative studies as it provides a context which would be helpful for the reader to understand and interpret the findings of this study. It would also be useful to specify the reason as to why Tanzania, and Dar es Salaam in particular was considered suitable for this study among other others.

-Line 145 “setting” please rephrase this to “study setting”

-In lines 146—148 (page 8) the authors state: “For each interview, the team decided in advance who would be most appropriate person to interview each stakeholder, matching wherever possible, the interviewer and stakeholder by gender, age, and other key characteristics.” In the methods section, would be good to specify the age of the interviewers involved in collecting data for this study, and clarify the potential effects of the interviewers’ age on the data collection.

-Lines 125-134: It is important for the authors to clarify in the “data collection and sampling” section about what guided the decisions on the sample size engaged in generation of data for this study. Was it guided by the principle of saturation? If so, what were the steps used in determining if the saturation was achieved? According to the authors, the prior actual number of participants in this study was anticipated to be 2-3 experts in each category, and indeed they interviewed 2 cultural leaders, 3 sexual experts, 2-community leaders, and 3 religious’ leaders. Can authors comment about this coincidence? Additionally, given the diversity of the categories of participants (sexual health experts, religious, community, and political leaders) engaged in this study, it would be useful if the authors could clarify if the saturation was reached for each type of participants. I can see some highlights on data saturation in the section about the limitation of the study (lines 560-561). I would encourage you to move this information earlier in the sampling sub-section.

-Lines 137-138: “This allowed the team to capture stakeholders’ experiences, and to identify the training needs of current students in each respective discipline.” Which students? I though the data reported in this paper are based on the stakeholders’ narratives on myths and conceptions about sexual health. Please clarify.

-Lines 143-144: The author state: “The interviewers were clinical faculty in medicine and midwifery from MUHAS…”. This sentence is confusing. Please check.

-Line 145: “All interviewers were bilingual in English and Kiswahili.” Please remove this information as it is already mentioned in lines 142-143.

-Lines 145-146: “...were female…” –Missing punctations (replace the word “female” with “females”)

-In lines 149-150, the authors state: “At the start of the interview, the stakeholder was invited to respond in whichever language they preferred”. Do you mean any language or you wanted to mean that the participants were given freedom to choose between Swahili and English languages? Please check and rectify.

-In the analysis section (Lines 168-169), the authors state: “A deductive-inductive coding strategy informed by grounded theory principles was employed to develop the codebook and code the transcripts.” As the author may be aware, unlike the inductive coding approach, the deductive coding is not informed by the grounded theory principles. Please correct that statement.

-Lines 189-190: “This paper reports the responses to the questions about common myths and misconceptions” Please add “about sexual health”.

-Lines 189-190: “Open coding involved reading several times three hard copy transcripts of the interviews and coding the interviews manually.” Given the diversity of the participants (i.e., religious, health experts etc.), I wonder if the tree transcripts reviewed for open coding were representative of the sub-populations involved in this study.

-Strongly advise that you include the codebook as a supporting material.

-Lines 562-563: “While the interviews were conducted in both Swahili and English, the scripts were translated to and analyzed in English.” It is important to also include this information in the data analysis section.

Results

Lines 287-293: The information mentioned here is about other contexts in Africa. This make me wonder whether the study explored about the myths and misconceptions around sexual health specific to Tanzania only or any other setting that that the participants knew about?

-Lines 411-423: “…First, both the community and health professionals may hold significant misgivings about…of the overall population.” Is the information presented in these lines informed by the study findings? They way it is presented now sounds like the authors’ assumptions.

Discussion

-The discussion section is generally good, but engagement with a broader range of qualitative literature in Tanzania around perceptions, beliefs and myths around sexual health would help. Put simply, how did the study findings align with other studies in Tanzania?

-The policy implications of the findings are not clear.

-I have concerns about some of the conclusions drawn from the manuscript, particularly the suggestion in lines 549-550 where the authors state: “We expect that after health care providers are trained in sexual health, they will begin to educate their patients during clinic visits and through community education”. Surely the authors—as they described in the introduction section—know that the cultural and social context informs health workers’ behaviors/practices, including those related to sexuality. For instance, it has been shown that in Tanzania (Mbekenga et al., 2013 https://doi.org/10.1186/1472-698X-13-4, Mchome et al., 2020 https://doi.org/10.1111/mcn.13048), despite representing the medical discourse, health care workers emerged as conveyors of the myths and misconceptions around sexuality and breastfeeding during postpartum period. These evidences suggest that awareness or knowledge messages will hardly shift strongly held cultural norms around sexuality. Thus, efforts more than education are needed to make health workers willing and courageous to talk about sexual issues / topics with their clients.

-I would suggest adding directions for future research at the end of the discussion so that others interested in this topic of research know how to use this manuscript in the future.

Good luck!

7. PLOS authors have the option to publish the peer review history of their article (what does this mean?). If published, this will include your full peer review and any attached files.

Reviewer #1: **Yes: **Kim Jonas, PhD

Reviewer #3: No

---

## [Author Response · Author response to Decision Letter 1]

3 Jan 2022

Dear Editor and Reviewers:

Thank you for your helpful comments and positive review of our manuscript entitled “Community Myths and Misconceptions about sexual health in Tanzania: Stakeholders' Views from qualitative study in Dar es Salaam Tanzania”. Below please find a point-by-point summary of the comments (in regular type) together with our response (in italics).

Reviewer 1

I'd like to thank the authors for attempting to address my and reveiwer2's previous comments. I am happy with this revision but have further suggestions...

Abstract:

the authors mentioned in their response letter that they opted for "stakeholders" but there's informants in line 34.

We apologize. We have replaced the synonyms (informants) with “stakeholders” as suggested by the reviewer throughout the paper. 

Introduction

Please check your referencing... only one box is needed to house the ref number... e.g., myths and misconceptions are common in Tanzania [11-15] and not [11] - [15], no, this is not how it goes... This comment applies throughout the manuscript in text referencing. If the references are not chronological you can still put them in one square box this way [5, 7,11-4] ... Hope this is making sense. Watch out for spacing before the in text refencing/ square boxes.

Thank you for this reminder. We confirm we have checked the referencing and have placed only one box to house the reference number/s as required. 

Methods

line 125, please rearrange to read: Sampling and data collection process

We agree, we have rearranged the subheading to read “Sampling and Data Collection Process” as advised by the reviewer. 

line 139, here's key informants again...? I expected to see the reference to the "Additional file" of the interview guide in this section but I cannot find it...It would be ideally places somewhere by line 148

Thank you for reminding us again, as described early we have replaced the other synonyms(informants) with “stakeholders” as suggested by the reviewer throughout the paper. And we have attached the interview guide as recommended at first mention of the guide.

Nicely described analysis process!

Thank you for your kind comment. 

Results

Table 2, page 14... there's a typo- .... unplanned pregnancies in the "world"

line 277, another "informant" and in line 327, 338, 348... there's more, please ensure you check this for consistency!

Thank you. We have corrected the typo, and replaced the word “informants” with “stakeholder” throughout the paper for consistency. 

Otherwise, this is a nice paper and timely for health providers.

Thank you again for your kind assessment and appreciation 

Reviewer 3: 

Thanks to Plos One for the opportunity to review this manuscript and to the authors for their interesting work on this important topic. Generally, the paper is well-written. The authors did a good job in addressing the comments raised in a previous round of review. However, there are some issues in the manuscript that need to be addressed before it can be considered for publication. I have outlined these concerns below and hope that my comments are helpful to the authors if they are given the opportunity to revise the paper:

Introduction

-The introduction is fairly written. But I wonder if in Tanzania there are no studies that addressed myths and misconceptions around sexual health. I would urge the authors to acknowledge the existing evidence and situate their study within it.

Thank you for your nice comment. Initially, in the introduction, we had only two references directly from Tanzania [References 1 and 14]. We have added several more. Reference number 2 is from WHO in which Tanzania is involved too; reference number 4 is a systematic review from many countries including from Tanzania. And we have added other studies from Tanzania to enrich the study [see reference numbers 10-12]. These now appear on Page 5.

-It would be good to add some lines stating the situation of sexual health problems in Tanzania and the related myths and misconceptions.

Thank you for the comment. A paragraph about situation of sexual health in Tanzania and related myths and misconception has been added on Page 5. 

Methods

-Lines 131-133, the authors state: “The study team identified health, cultural, political and religious leaders who were known to deal with sexual health issues”. It is not clearly described who helped the researchers in identifying and approaching these key informants to participate in the interviews.

Participants of these study were selected purposively, the authors who live and work in Tanzania, together with our colleagues form the US met in January 2019 to identify who would be the most appropriate stakeholders to interview. Our dear late Dr. Leshabari who was co-Principal Investigator and senior faculty at MUHAS was the primary decision maker. All the stakeholders selected by our team are public figures known to deal with sexual health publicly via several media. The process is clearly described in last paragraph of page 7.

-Under the sub-section “Setting”, line 145-146, the authors state: “Data were collected from June to August 2019 in Dar es Salaam, Tanzania.” I think the authors should provide further description of the setting of this study. This is particularly important for qualitative studies as it provides a context which would be helpful for the reader to understand and interpret the findings of this study. It would also be useful to specify the reason as to why Tanzania, and Dar es Salaam in particular was considered suitable for this study among other others.

Thank you for this comment. We have described the reason why Dar es salaam is a study area for this study. Please see page 7.

-Line 145 “setting” please rephrase this to “study setting”

Thank you, we have rephrased it to “Study Setting.” 

-In lines 146—148 (page 8) the authors state: “For each interview, the team decided in advance who would be most appropriate person to interview each stakeholder, matching wherever possible, the interviewer and stakeholder by gender, age, and other key characteristics.” In the methods section, would be good to specify the age of the interviewers involved in collecting data for this study, and clarify the potential effects of the interviewers’ age on the data collection.

Age range of clinical faculty who conducted the interview ranged from 35 to 60 years. See page 8

The potential effects of the interviewers’ age on the data collection

This allows stakeholders(interviewee) and interviewer to be more confrontable during the interview and share what they know related to sexual health. See page 8

-Lines 125-134: It is important for the authors to clarify in the “data collection and sampling” section about what guided the decisions on the sample size engaged in generation of data for this study. Was it guided by the principle of saturation? If so, what were the steps used in determining if the saturation was achieved? According to the authors, the prior actual number of participants in this study was anticipated to be 2-3 experts in each category, and indeed they interviewed 2 cultural leaders, 3 sexual experts, 2-community leaders, and 3 religious’ leaders. Can authors comment about this coincidence? Additionally, given the diversity of the categories of participants (sexual health experts, religious, community, and political leaders) engaged in this study, it would be useful if the authors could clarify if the saturation was reached for each type of participants. I can see some highlights on data saturation in the section about the limitation of the study (lines 560-561). I would encourage you to move this information earlier in the sampling sub-section.

We confirm our study team included a qualitative expert methodologist who trained all members in qualitative methods including the goal of saturation. Our goal was to have 2-3 experts in each category participate so as to attain overall saturation, and indeed we had 2 cultural leaders, 3 sexual experts, 2-community leaders, and 3 religious’ leaders in which we reached saturation from the information we got from them. The exception was politicians. We tried to recruit multiple politicians but were only able to recruit one who was willing to discuss sexual health and key populations. So, in that category, we were unable to confirm if saturation as reached or not. This information is now included as a limitation in the limitations section. We have also included this information in the methodology section. Please see page 9. 

-Lines 137-138: “This allowed the team to capture stakeholders’ experiences, and to identify the training needs of current students in each respective discipline.” Which students? I though the data reported in this paper are based on the stakeholders’ narratives on myths and conceptions about sexual health. Please clarify.

It’s important to understand that the information obtained from this study had two objectives. As researchers, we wanted to document the common myths and misconceptions related to sexual health in Tanzania. More immediately, we wanted to identify community myths to tailor a sexual heath curriculum for health students in Tanzania. We confirm we have integrated the findings from this study into the curriculum which is currently being evaluated. 

-Lines 143-144: The author state: “The interviewers were clinical faculty in medicine and midwifery from MUHAS…”. This sentence is confusing. Please check.

All interviewers were faculty members (lecturers) from Muhimbili University of Health and Allied Sciences. Two of them were midwives teaching in the School of Nursing, and one was a medical doctor teaching in the School of Medicine. As clinical faculty, they all are also clinicians working at the Muhimbili National hospital. 

-Line 145: “All interviewers were bilingual in English and Kiswahili.” Please remove this information as it is already mentioned in lines 142-143.

Thank you for observation. We have removed the repeated line as suggested by reviewer. Please see page 8.

-Lines 145-146: “...were female…” –Missing punctations (replace the word “female” with “females”)

Thank you for observation. We have corrected that.

-In lines 149-150, the authors state: “At the start of the interview, the stakeholder was invited to respond in whichever language they preferred”. Do you mean any language or you wanted to mean that the participants were given freedom to choose between Swahili and English languages? Please check and rectify.

Again, thank you for observation. We have changed the sentence to read “At the start of the interview, the stakeholder was invited to respond in either English or Swahili, with the most preferring to use English.”

-In the analysis section (Lines 168-169), the authors state: “A deductive-inductive coding strategy informed by grounded theory principles was employed to develop the codebook and code the transcripts.” As the author may be aware, unlike the inductive coding approach, the deductive coding is not informed by the grounded theory principles. Please correct that statement.

Thank you for the reminder. We have corrected that statement. Please see page 9.

-Lines 189-190: “This paper reports the responses to the questions about common myths and misconceptions” Please add “about sexual health”.

Thank you. We have added the word about sexual health, now the sentence read. “This paper reports the responses to the questions about common myths and misconceptions about sexual health”

-Lines 189-190: “Open coding involved reading several times three hard copy transcripts of the interviews and coding the interviews manually.” Given the diversity of the participants (i.e., religious, health experts etc.), I wonder if the tree transcripts reviewed for open coding were representative of the sub-populations involved in this study.

Thank you for an observation. We apologize for the confusion. We read one transcript several time for each subpopulation. Therefore, we read four transcripts and coded them manually in this initial phase to determine early ideas and emerging themes by using an open-ended coding format. We have also deleted the word three and put four in revised manuscript for clarity. Please see page 10. 

-Strongly advise that you include the codebook as a supporting material.

Thank you, the codebook has been added as a supporting material as suggested by the reviewer.

-Lines 562-563: “While the interviews were conducted in both Swahili and English, the scripts were translated to and analyzed in English.” It is important to also include this information in the data analysis section.

We agree. This detail now appears in line 218 of the Data Management and Analysis section. 

Results

Lines 287-293: The information mentioned here is about other contexts in Africa. This make me wonder whether the study explored about the myths and misconceptions around sexual health specific to Tanzania only or any other setting that that the participants knew about?

We were clear with stakeholders that we were talking about common myths and misconceptions in Tanzania. In their responses, some stakeholders identified a myth or misconception which they said was not limited to Tanzania only, but also was common in other African countries. That is why we presented it in that way. 

-Lines 411-423: “…First, both the community and health professionals may hold significant misgivings about…of the overall population.” Is the information presented in these lines informed by the study findings? The way it is presented now sounds like the authors’ assumptions.

We confirm it is informed by the study’s findings. Specifically, stakeholders referenced that the community is concerned that health care providers who provide services to key populations (i.e., men who have sex with men, sex workers) are, in fact, promoting or endorsing such behavior. 

Discussion

-The discussion section is generally good, but engagement with a broader range of qualitative literature in Tanzania around perceptions, beliefs and myths around sexual health would help. Put simply, how did the study findings align with other studies in Tanzania?

We have added a paragraph relating to findings of this study with similar findings from prior research in Tanzania. 

-The policy implications of the findings are not clear.

We agree. As we hope we have made clear, our aim in this study was to inform and tailor a sexual health training curriculum for health students in Dar es Salaam. It was not to inform policy. 

-I have concerns about some of the conclusions drawn from the manuscript, particularly the suggestion in lines 549-550 where the authors state: “We expect that after health care providers are trained in sexual health, they will begin to educate their patients during clinic visits and through community education”. Surely the authors—as they described in the introduction section—know that the cultural and social context informs health workers’ behaviors/practices, including those related to sexuality. For instance, it has been shown that in Tanzania (Mbekenga et al., 2013 https://doi.org/10.1186/1472-698X-13-4, Mchome et al., 2020 https://doi.org/10.1111/mcn.13048), despite representing the medical discourse, health care workers emerged as conveyors of the myths and misconceptions around sexuality and breastfeeding during postpartum period. These evidences suggest that awareness or knowledge messages will hardly shift strongly held cultural norms around sexuality. Thus, efforts more than education are needed to make health workers willing and courageous to talk about sexual issues / topics with their clients.

We respectfully agree in part and disagree in part. 

Here’s where we agree. In our focus groups with 121 providers and healthcare students (not reported here), we heard several accounts where, in the absence of sexual health education and sexual counseling skills development, medical doctors, nurses and midwives held similar myths and misconceptions around sexual health as the community. For example, they worried that providing a sexually active 14-year-old girl with contraception could encourage sexual promiscuity, that treating HIV or STIs in a man who has sex with men might encourage homosexuality, or that reporting a man for raping or beating his wife might have negative effects on the marriage. We note our findings are consistent with Mbekenga et al.’s observation that “health care workers were sometimes described as conveyors.” But Mbekenga et al. emphasizes “sometimes,” and we also appreciate their caveat that “public health services were not discussed much in the FGDs.” We note the Mchome et al. (2020) study was conducted in a single rural village in Kilosa where there is no health facility, and no doctors, nurses or midwives. Instead, the study interviewed birth attendants and community health workers who are less trained and therefore possibly more likely to convey myths and misconceptions. 

Where we disagree is in the conclusion that awareness or knowledge will hardly shift strong held cultural norms around sexuality. We respectfully would suggest that it is too sweeping a generalization. And it is not based in evidence. In both the pilot we conducted and the randomized controlled trial of the tailored sexual health curriculum currently in progress at MUHAS, we are seeing large shifts in the knowledge and attitudes of medical, nursing and midwifery students who received sexual health education compared to the waitlist control group (data still under analysis). This suggests that a strong curriculum can significantly modify health students’ attitudes, beliefs, and counseling skills in Tanzania. It may not be perfect, but our students are strongly committed to practicing evidence-informed healthcare. It seems, if you provide students in Tanzanian with sexual health education, many/most will modify their beliefs accordingly. 

-I would suggest adding directions for future research at the end of the discussion so that others interested in this topic of research know how to use this manuscript in the future.

We agree and have added a paragraph on this.

Thank you to the reviewers for their helpful feedback.

Best regards,

Gift Lukumay, BScN, MSc.PHEC

Department of Community Health Nursing 

Muhimbili University of Health and Allied Sciences

---

## [Decision Letter · Decision Letter 2]

31 Jan 2022

PONE-D-20-40972R2Community myths and misconceptions about sexual health in Tanzania: stakeholders’ views from a qualitative study in Dar es Salaam Tanzania.PLOS ONE

Dear Dr. %Rosser %,

Thank you for submitting your manuscript to PLOS ONE. After careful consideration, we feel that it has merit but does not fully meet PLOS ONE’s publication criteria as it currently stands. Therefore, we invite you to submit a revised version of the manuscript that addresses the points raised during the review process.

 Please address the pints made by reviewer #2. Please ensure that your decision is justified on PLOS ONE’s publication criteria and not, for example, on novelty

We look forward to receiving your revised manuscript.

Kind regards,

Mary Hamer Hodges, MBBS MRCP DSc

Academic Editor

PLOS ONE

Journal Requirements:

Additional Editor Comments (if provided):

Please clarify the points raised by reviewer #2.

Reviewers' comments:

Reviewer's Responses to Questions

**Comments to the Author**

1. If the authors have adequately addressed your comments raised in a previous round of review and you feel that this manuscript is now acceptable for publication, you may indicate that here to bypass the “Comments to the Author” section, enter your conflict of interest statement in the “Confidential to Editor” section, and submit your "Accept" recommendation.

Reviewer #1: All comments have been addressed

Reviewer #3: (No Response)

2. Is the manuscript technically sound, and do the data support the conclusions?

Reviewer #1: Yes

Reviewer #3: Yes

3. Has the statistical analysis been performed appropriately and rigorously? 

Reviewer #1: N/A

Reviewer #3: N/A

4. Have the authors made all data underlying the findings in their manuscript fully available?

Reviewer #1: Yes

Reviewer #3: (No Response)

5. Is the manuscript presented in an intelligible fashion and written in standard English?

Reviewer #1: (No Response)

Reviewer #3: Yes

6. Review Comments to the Author

Reviewer #1: No further comments. well done on addressing the previous comments and those of the other reviewer satisfactorily.

Reviewer #3: I thank the authors for addressing my previous comments. I am pleased with the revision but have one more observation, particularly on data analysis section.

In line 187-192: “Coding approach”. The authors state “An inductive coding strategy informed by grounded theory principles was employed to develop the codebook and code the transcripts.” Which implies that only an inductive approach was engaged in developing codes. Yet, in line 201-203, the author state “Both deductive and inductive codes continued to be generated and added to the codebook iteratively during the three phases of coding.” Did the analysis only engage the inductive approach? If so, the statement in lines 201-203 is irrelevant. In case the analysis engaged both inductive and deductive strategies in developing codes and coding the data, it is important that in lines 187-192 where the authors talk about the coding approach, they also mention the deductive approach in addition to the inductive one. For example, a statement like, “The inductive and deductive strategies were employed to develop the codebook and code the transcripts. First, a series of inductive codes was developed based on the principles of Grounded Theory. Second, the deductive coding was performed based on xxx.”.

Good luck!

7. PLOS authors have the option to publish the peer review history of their article (what does this mean?). If published, this will include your full peer review and any attached files.

Reviewer #1: **Yes: **Kim Jonas, PhD

Reviewer #3: No

---

## [Author Response · Author response to Decision Letter 2]

8 Feb 2022

Dear Editor and Reviewers:

Thank you for your email query regarding our manuscript entitled “Community Myths and Misconceptions about sexual health in Tanzania: Stakeholders' Views from qualitative study in Dar es Salaam Tanzania”. Below please find the comment from the reviewer (in regular type) followed by our response (in italics).

Reviewer 3 

I thank the authors for addressing my previous comments. I am pleased with the revision but have one more observation, particularly on data analysis section.

In line 187-192: “Coding approach”. The authors state “An inductive coding strategy informed by grounded theory principles was employed to develop the codebook and code the transcripts.” Which implies that only an inductive approach was engaged in developing codes. Yet, in line 201-203, the author state “Both deductive and inductive codes continued to be generated and added to the codebook iteratively during the three phases of coding.” Did the analysis only engage the inductive approach? If so, the statement in lines 201-203 is irrelevant. In case the analysis engaged both inductive and deductive strategies in developing codes and coding the data, it is important that in lines 187-192 where the authors talk about the coding approach, they also mention the deductive approach in addition to the inductive one. For example, a statement like, “The inductive and deductive strategies were employed to develop the codebook and code the transcripts. First, a series of inductive codes was developed based on the principles of Grounded Theory. Second, the deductive coding was performed based on xxx.”.

Thank you for pointing out this inconsistency. We confirm we used deductive and inductive coding approach in our analysis. As recommended by the reviewer, we have added “and deductive” in lines 187-192. Hopefully, this clears up the confusion (see page 10).

Thank you to the reviewer for their helpful feedback.

Best regards,

Gift Lukumay, BScN, MSc.PHEC

Department of Community Health Nursing 

Muhimbili University of Health and Allied Sciences

---

## [Decision Letter · Decision Letter 3]

16 Feb 2022

Community myths and misconceptions about sexual health in Tanzania: stakeholders’ views from a qualitative study in Dar es Salaam Tanzania.

PONE-D-20-40972R3

Dear Dr. %Rosser%,

We’re pleased to inform you that your manuscript has been judged scientifically suitable for publication and will be formally accepted for publication once it meets all outstanding technical requirements.

Kind regards,

Mary Hamer Hodges, MBBS MRCP DSc

Academic Editor

PLOS ONE

Additional Editor Comments (optional):

Reviewers' comments:

Reviewer's Responses to Questions

**Comments to the Author**

1. If the authors have adequately addressed your comments raised in a previous round of review and you feel that this manuscript is now acceptable for publication, you may indicate that here to bypass the “Comments to the Author” section, enter your conflict of interest statement in the “Confidential to Editor” section, and submit your "Accept" recommendation.

Reviewer #3: All comments have been addressed

2. Is the manuscript technically sound, and do the data support the conclusions?

Reviewer #3: Yes

3. Has the statistical analysis been performed appropriately and rigorously? 

Reviewer #3: N/A

4. Have the authors made all data underlying the findings in their manuscript fully available?

Reviewer #3: (No Response)

5. Is the manuscript presented in an intelligible fashion and written in standard English?

Reviewer #3: Yes

6. Review Comments to the Author

Reviewer #3: (No Response)

7. PLOS authors have the option to publish the peer review history of their article (what does this mean?). If published, this will include your full peer review and any attached files.

Reviewer #3: No

---

## [Editor Report · Acceptance letter]

5 Apr 2022

PONE-D-20-40972R3 

Community myths and misconceptions about sexual health in Tanzania: stakeholders’ views from a qualitative study in Dar es Salaam Tanzania. 

Dear Dr. Rosser:

I'm pleased to inform you that your manuscript has been deemed suitable for publication in PLOS ONE. Congratulations! Your manuscript is now with our production department. 

Kind regards, 

on behalf of

Prof. Mary Hamer Hodges 

Academic Editor

PLOS ONE